# Formulation of Bioerodible Ketamine Microparticles as an Analgesic Adjuvant Treatment Produced by Supercritical Fluid Polymer Encapsulation

**DOI:** 10.3390/pharmaceutics10040264

**Published:** 2018-12-06

**Authors:** Felicity Y. Han, Andrew K. Whittaker, Steven M. Howdle, Andrew Naylor, Anjumn Shabir-Ahmed, Cheng Zhang, Maree T. Smith

**Affiliations:** 1School of Biomedical Sciences, Faculty of Medicine, The University of Queensland, Brisbane QLD 4072, Australia; f.han@uq.edu.au; 2Australian Institute for Bioengineering and Nanotechnology, The University of Queensland, Brisbane QLD 4072, Australia; a.whittaker@uq.edu.au (A.K.W.); c.zhang3@uq.edu.au (C.Z.); 3ARC Centre of Excellence in Convergent Bio Nano Science and Technology, The University of Queensland, Brisbane QLD 4072, Australia; 4School of Chemistry, University of Nottingham, Nottingham NG7 2RD, UK; Steve.Howdle@nottingham.ac.uk; 5Critical Pharmaceuticals Ltd., BioCity Nottingham, Nottingham NG1 1GF, UK; anjumnshabir@yahoo.com; 6Upperton Limited, Biocity Nottingham, Nottingham NG7 2TN, UK; anaylor@upperton.com; 7School of Pharmacy, Faculty of Health and Behavioural Sciences, The University of Queensland, Brisbane QLD 4072, Australia

**Keywords:** analgesic adjuvant, ketamine, cancer pain, drug delivery, poly(lactic-*co*-glycolic acid) (PLGA), sustained release

## Abstract

Pain is inadequately relieved by escalating doses of a strong opioid analgesic such as morphine in up to 25% of patients with cancer-related severe pain complicated by a neuropathic (nerve damage) component. Hence, there is an unmet medical need for research on novel painkiller strategies. In the present work, we used supercritical fluid polymer encapsulation to develop sustained-release poly(lactic-*co*-glycolic acid) (PLGA) biodegradable microparticles containing the analgesic adjuvant drug ketamine, for injection by the intrathecal route. Using this approach with a range of PLGA co-polymers, drug loading was in the range 10–60%, with encapsulation efficiency (EE) of 60–100%. Particles were mainly in the size range 20–45 µm and were produced in the absence of organic solvents and surfactants/emulsifiers. Investigation of the ketamine release profiles from these PLGA-based microparticles in vitro showed that release took place over varying periods in the range 0.5–4.0 weeks. Of the polymers assessed, the ester end-capped PLGA5050DLG-1.5E gave the best-controlled release profile with drug loading at 10%.

## 1. Introduction

Unrelenting severe cancer-related pain, particularly that complicated by a neuropathic (nerve damage) component, is difficult to alleviate by escalating doses of strong opioid analgesics such as morphine given alone or in combination with adjuvant analgesic agents by conventional systemic dosing routes [1]. Although cancer can be a terminal disease, there should be no reason to deny a patient the opportunity to live productively and free of pain [2]. Therefore, the goals of pain control should be to optimize the patient’s comfort and function whilst avoiding unnecessary adverse effects from medications [3]. 

The intrathecal (i.t.) dosing route enables logarithmic scale reductions in the analgesic/adjuvant doses needed to produce adequate pain relief because this route bypasses the systemic metabolism and delivers the drugs near to the receptors/ion channels that transduce pain relief in the spinal cord [4,5]. Importantly, there is some evidence that, in the postoperative setting, adding ketamine to an opioid in patient-controlled analgesia has a beneficial effect on pain relief, is morphine-sparing, and produces less postoperative nausea and vomiting compared with equi-effective opioid analgesic administration alone [6,7]. The efficacy of ketamine alone as an analgesic is still under debate, especially for indications such as chronic pain [8]. When used as an analgesic, sub-anaesthetic doses of ketamine are administered [9]. In rodents, co-injection of ketamine with a strong opioid analgesic attenuated the development of analgesic tolerance and enhanced pain relief [10]. Together, these findings suggest that i.t. co-administration of a strong opioid analgesic with low-dose ketamine in close proximity to their target receptors/ion channels in the spinal cord may produce satisfactory analgesia. 

As implantable i.t. delivery devices are associated with a range of catheter-related problems in up to 25% of patients [11] and the surgery is highly invasive, it is clear that new therapeutic approaches are needed. One such strategy is the development of sustained-release biodegradable microparticles for i.t. injection with the potential to restore satisfactory analgesia in patients who would otherwise suffer from intractable cancer-related pain despite the administration of escalating doses of a strong opioid analgesic by conventional oral or parenteral dosing routes [12]. However, successful microencapsulation of analgesic/adjuvant drugs of interest must overcome the challenges of poor drug loading, unacceptably high initial burst release, inadequate sustained-release period, and contamination by unacceptable organic solvent residues.

Polymeric particles are widely used in the field of drug delivery [12,13,14]. The most appropriate carriers for sustained drug delivery are slowly degrading polymers. For example, poly(lactic-*co*-glycolic) acid (PLGA) and poly(lactic acid) (PLA) are biocompatible, biodegradable polymers that have been approved by the United States Food and Drug Administration (FDA) and the European Medicines Agency (EMA) for use in systemic drug delivery systems [13]. Of these, PLGA-based drug-loaded microparticles, commonly prepared by a water-in-oil-in-water (w/o/w) double-emulsion method or an oil-in-water (o/w) single-emulsion method, hold promise because of their biodegradability and successful application in a range of systemically administered drug formulations [14,15]. 

With the traditional methods of sustained-release microparticle formulation, large quantities of potentially toxic organic solvents and/or surfactants/emulsifiers are used, possibly leading to unacceptable levels of residual impurities that necessitate the use of further purification steps to ensure their removal [16]. In contrast, supercritical CO_2_ (scCO_2_) technology provides a means to prepare particles that are environmentally benign and produce particles that are free from organic solvent residues after processing [17]. The method is flexible and allows control of the composition of the polymer mixtures to control the rates of release of the encapsulated molecules [18]. In particular, this technology provides a clean and reliable method for the production of particles containing therapeutic molecules such as ketamine. 

The use of scCO_2_ to create drug-loaded particles is of particular interest [19] because of its relatively mild conditions (7.4 MPa and 31 °C), allowing the processing of thermally sensitive substances [20,21]. By varying the temperature and pressure, the properties of scCO_2_ can be changed, enabling it to act as a solvent, antisolvent, solute, or drying agent [22]. Consequently, different particle characteristics can be designed by adjusting the scCO_2_ processing conditions including temperature, pressure, contact time, diffusion processes, and depressurization conditions [23,24,25]. 

The precise method of processing using scCO_2_ depends upon the respective solubilities of the polymer carrier and the drug in the solvent (scCO_2_) [26]. Howdle and his team reported a method for the production of hGH-loaded polyester particles and demonstrated controlled release in a simian model [27]. In other work, vaccines for tetanus toxoid (TT) were incorporated into PLA particles using scCO_2_ processing [28]. A single injection of the polymer-TT particles into mice produced similar antibody titres as multiple injections of a commercial alum-adsorbed TT vaccine [28]. These results demonstrate the potential for the preparation of organic solvent-free particles loaded with the molecules of interest.

The simple CriticalMix process utilizes scCO_2_ to encapsulate active pharmaceutical ingredients (APIs) within a biodegradable polymer matrix for sustained-release applications [26,27,29]. Hence, the aim of our present study was to microencapsulate ketamine into biodegradable polymers to produce sustained-release microparticle formulations for potential i.t. co-injection with strong opioid analgesic-loaded microparticles so as to potentially achieve prolonged periods of satisfactory pain relief. 

## 2. Materials and Methods

### 2.1. Materials

Various poly(lactic-*co*-glycolic acid) (PLGA) biodegradable polymers (50:50), (75:25), including 5050DLG-1.5E (5050-1.5E), RG503, RG504, RG7525S, 9010DLG-4.5E (9010-4.5E), 5050DLG-1A (5050-1A), 5050DLG-2A (5050-2A), 5050DLG-4A (5050-4A), 7525DLG-4A (5050-4A), and poly(lactic-acid) (PLA)-2.5A and R202S, were purchased from Evonik Health care (Birmingham, Al, USA). The physicochemical characteristics of these polymers are listed in Table 1. Ketamine HCl powder (98–99%) was from Sigma-Aldrich (Haverhill, UK). Food-grade CO_2_ was from BOC (Rotherham, UK).

### 2.2. Methods

#### 2.2.1. Preparation of Ketamine-Loaded Polymeric Microparticles

Firstly, ketamine HCl was converted to the corresponding free base by adjusting the pH to ~10 using 2 M NaOH added dropwise according to our previous work (Han et al., 2015). The CritcalMix scCO_2_ process was optimised on the basis of that described previously [29]. For each of the polymer powders listed in Table 1, a weighed amount was mixed with ketamine in varying ratios (Table 2), and each mixture was loaded into a pressure vessel, pressurised with CO_2_ and heated to 40 °C at a pressure of 14 MPa. Once the desired temperature was reached, the mixture of CO_2_/polymer/drug inside the vessel was mixed using mechanical stirring. Liquefaction of the polymer by scCO_2_ enabled the drug to be mixed into the polymer. Approximately 0.5–1.0 h later heating was stopped, and the polymer/ketamine/scCO_2_ mixture was allowed to cool to below 25 °C before the CO_2_ was slowly vented, depressurised through a disk nozzle with a 0.6 mm orifice (unheated), and the polymer solidified to entrap ketamine and form microparticles containing ketamine. Cooling took place over a period of one hour with pressure decreasing from 14 MPa scCO_2_ to 7.5 MPa (~1000 psi). The microparticles were collected and ground with a pestle and mortar and then sieved through a 45 µm sieve. The final products were stored refrigerated at a mean (±SD) temperature of 5(±3) °C and protected from light in a desiccator.

#### 2.2.2. Determination of Drug Incorporation Efficiency

Accurately weighed amounts (~10 mg; *n* = 3) of the ketamine-loaded polymeric microparticles (Table 1) were dissolved in acetonitrile. The ketamine concentrations were quantified using HPLC (Agilent 1200 series) with UV detection at 280 nm. A gradient HPLC method was used for determining the drug loading (DL) using a Waters XBridge BEH C18 column: 3.5 µm, 3.0 × 150 mm. The calibration range was 5–800 µg/mL. The mean (±SD) correlation coefficients for ketamine calibration curves were 0.9998 (±0.0001). 

Drug incorporation efficiency, expressed as actual drug loading (% w/w), and encapsulation efficiency (EE % w/w) were calculated using Equations (1) and (2), respectively. The individual values for three replicate determinations and their mean values are reported.

Drug loading (%) = 100 × mass of drug in microparticles/weight of microparticles(1)

EE (%) = 100 × mass of drug in microparticles/mass of drug in microparticles theoretically(2)

#### 2.2.3. Determination of In Vitro Drug Release

Accurately weighed ~20 mg samples of ketamine-loaded polymeric microparticles (*n* = 3) were suspended in 1mL of PBS and transferred to dialysis tubes (SnakeSkin™ Dialysis Tubing, 3.5K MWCO). Each dialysis tube was sealed, placed into a capped container containing 20mL PBS, and then placed into an incubator maintained at 37.5 °C and shaken horizontally at an oscillating frequency of 120 min^−1^. Sampling time points were 3 and 24 h, 3, 7, 14, 21, and 28 days. At each time point, a 1 mL aliquot of buffer was taken for analysis and replaced with fresh buffer. The 1 mL samples (*n* = 3) were filtered using a 0.2 µm nylon syringe filter into a vial for HPLC analysis of the ketamine concentrations.

An isocratic HPLC method was optimised for determining the ketamine release profile using a C18 column: 3.5 µm, 3.0 × 150 mm. The calibration range was 5–800 µg/mL. The mean (±SD) correlation coefficients for ketamine calibration curves were *R*^2^ > 0.9999. 

### 2.3. Morphology and Particle Size

Morphological evaluation of the sustained-release ketamine-loaded PLGA microparticles was performed using scanning electron microscopy (Jeol IT300, JEOL Ltd., Tokyo, Japan) to determine shape and surface morphology. The microparticles were sputter-coated with platinum using an Auto Smart Coater (JFC-1300, JEOL Ltd., Tokyo, Japan) before examination using scanning electron microscopy. 

### 2.4. Data Analysis

The data are presented as mean ± standard deviation (SD) or standard error of the mean (SEM) (±SD/SEM). Non-linear regression was used to calculate in vitro drug release using GraphPad Prism^TM^ v7.03 (GraphPad, San Diego, CA, USA). The one-way and two-way ANOVA followed by the Bonferroni test were used for comparing the means of burst release for the various formulation trials herein. The statistical significance criterion was *p* ≤ 0.05. To investigate the ketamine release kinetics, in vitro drug release data were fitted to various kinetic models, and regression analysis was performed for the selected formulations. The data were plotted as cumulative percentage drug release versus time for the zero-order model, square root of time for the Higuchi model, and log time for the Korsmeyer–Peppas model [30,31,32]. 

## 3. Results

### 3.1. Successful Encapsulation of Ketamine

A total of 112 trials with various PLGA co-polymers individually, as PLGA blends, or as PLGA blended with PLA were conducted. Our focus was upon minimisation of the burst and optimisation of the release profile over 24 h. Thus, we varied temperature, pressure, contact time, stirring rate of the scCO_2_/polymer/drug solution, as well as depressurization conditions including CO_2_ venting rate and venting temperature to deliver the optimal formulation. Ketamine as a free base is stable under supercritical CO_2_ conditions, as no degradation was observed when processed under CriticalMix conditions (data not shown). Ketamine loading was controlled in the range of 10–60% (Figure 1 and Table 2). The recovery of ketamine from the microparticle formulations was in the range 60–100% (mean of 80%) for all trials. Details of the selected 19 formulations of ketamine-loaded microparticles produced herein are summarized in Table 2, including the mean (±SD) actual drug loading (*n* = 3) and EE.

### 3.2. In Vitro Release Profiles

For all formulations of the ketamine-loaded microparticles there was a higher initial release phase when the drug loading was increased (Figure 1), followed by a slower nearly zero-order release lasting for up to 28 days (Figure 2, Figure 3 and Figure 4). 

#### 3.2.1. Reducing the Ketamine Payload Reduced the Burst Release of Ketamine

When the ketamine payload exceeded 60%, all the ketamine-loaded microparticles formulated with various polymer compositions (high MW vs. low MW) performed similarly in vitro. The 24 h burst release values were over 70% for all trials (e.g., formulations (F) 1–4 in Figure 1). Therefore, the ketamine payload was reduced to 30%, which enabled a greater portion of the microparticle composition to be polymer-controlled and resulted in a reduction of the burst release of ~40% (from ~70% down to ~30%) (Figure 1 and Figure 2). A 14-day sustained release profile was achieved (Figure 2) for F5 (30% ketamine +70% PLGA5050-1.5E) together with a significantly reduced 24 h burst release (****, *p* < 0.0001) when compared with the corresponding parameters for F2 (60% ketamine + 40% PLGA5050-1.5E) using the same polymer PLGA5050-1.5E (Figure 2).

For microparticles formed using the polymer PLGA5050-1.5E, a further reduction in the ketamine loading from 30% to 10% enabled the polymer matrix to have better control over the release profile (Figure 2). Sustained release was extended for more than 21 days for F6 (10% ketamine + 90% PLGA5050-1.5E), and the 24 h burst release was reduced significantly (*p* < 0.0001) when loading was reduced from ~30% to ~10% compared with F5 (30% ketamine + 70% PLGA5050-1.5E). The release profile for microparticles loaded with 20% ketamine (F7, 20% ketamine + 80% PLGA5050-1.5E) did not differ significantly when compared with 30% ketamine-loaded microparticles (F5) (*p* > 0.05) (Figure 2), although the 24 h burst release was reduced from 27.7% to 17.5%. 

#### 3.2.2. Ester End-Capped Polymers Are Better at Controlling the Release of Ketamine Compared with Acid End-Capped Polymers

The polymers used in this study were PLGAs and PLAs with varying molecular weights (MW) and end-group functionalities. The physicochemical properties of the polymers used in this study are summarized in Table 1. Particles formulated using acid end-capped polymers had a high burst release, that is, ~55% release within the initial 24 h and >80% release within three days (Figure 3). Together, our data show that for acid end-capped PLGAs, changing the MW/viscosity and possibly the particle matrix had little influence on the rate of release. Our data show a maximum of one-week release in vitro with acid end-capped PLGA polymers (Figure 3). Additionally, varying the ratios of the PLGA5050-1A and the PLGA7525-4A polymers had little influence on the release profiles. High-MW acid end-capped polymers did not perform significantly better than low-MW acid end-capped polymers. Incorporation of a low-MW ester end-capped polymer PLGA5050-1.5E (50%) into PLGA7525-4A (20%) reduced the rate of ketamine release and extended the release over seven days (F8 in Figure 4). Ester end-capped polymers enabled ketamine-loaded microparticles to be formed with release extended to 14 days and more than 21 days (F5, F6, in Figure 2).

#### 3.2.3. In Vitro Drug Release Kinetics

For the kinetic studies, the following plots were made: % drug release versus time (zero-order kinetic model); % drug release versus square root of time (Higuchi model); log % drug release versus log time (Korsmeyer–Peppas model). All plots are given in the Appendix A, and the results are summarized in Table 3. Most data fit well to the Higuchi model and the Korsmeyer–Peppas model (*R*^2^ > 0.9000). The PLGA5050-1.5E-based F6 of 10% DL showed the best fit (*R*^2^ = 0.9481) of the observed data to the zero-order model.

### 3.3. Morphology and Particle Size

Visual inspection of the scanning electron micrograph images of F6 (the best performing sustained-release profile herein) in Figure 5 show that the microparticles were random in shape without significant aggregation and/or porosity and were in the size range of 20–45 μm. There was no clear change in particle morphology after six months storage at 2–8 °C (Figure 5B). 

## 4. Discussion

In the present study, we demonstrate for the first time that the microencapsulation of ketamine using supercritical CO_2_ technology resulted in microparticles that were solvent- and surfactant-free once CO_2_ was converted from the liquid to the gaseous state and vented to the atmosphere [22]. Carbon dioxide is relatively inert, non-toxic, and environmentally friendly when compared with organic solvents and emulsifying agents used in classical double-emulsion microparticle formulation methods [16]. As CO_2_ has a relatively low critical point of 7.38 MPa at 31.1 °C, drug-loaded microparticles can be processed under ambient conditions [33]. Our ketamine data herein highlight the applicability of scCO_2_ for the incorporation of small bioactive molecules into a range of polymer matrices to achieve the desired release profiles. Importantly, ketamine loading of these polymeric microparticle formulations could be controlled in the range of 10–60%, with EE in the range of 60–100%. For microparticles with ketamine loading in the range of 10–30%, the microencapsulated payloads were 1.5–4.0-fold higher (Table 2) than those achieved in our previous work (6.8%) using a w/o/w double emulsion method (Han et al., 2015). The particle sizes were mainly 20–45 µm (Figure 5), which was within our target size range of 20–60 µm (Han et al., 2015), and in vitro ketamine release was observed over a period of 0.5–4.0 weeks (Figure 2, Figure 3 and Figure 4). 

A greater proportion of polymer than encapsulated payload is required to achieve prolonged release. Our findings show that for ketamine, microparticle loading up to a maximum of 30% produced prolonged-release microparticles. Reducing the ketamine payload from 60% to 30% and then to 10% progressively reduced the burst release effect. However, on the flip side, reducing the encapsulated drug payload might adversely affect the requirement of achieving a therapeutic response in vivo (60–100 µg per day), but this remains to be assessed. Our finding that the ketamine release profile was prolonged by increasing the percentage of polymer in the formulation is aligned with work by others [34] whereby the polymer/encapsulated drug ratio was an important factor in controlling the rate of release. Herein, we found that to achieve prolonged release over 21 to 28 days, the polymer-to-microencapsulated ketamine ratio should be 70:30 or 90:10. 

Although higher drug loading led to more extensive burst release, the level of drug loading had little effect on the release rate after 24 h, even when the drug loading was as high as 60%. Drug release from PLGA microparticles is underpinned by three factors, namely, external diffusion, internal mass transfer, and polymer degradation [35]. On the basis of our present results, the external diffusion and internal mass transfer likely determine the extent of ketamine release during the first 24 h, and polymer degradation is the main factor controlling release thereafter. Our data extend the findings of Cai et al. [36] and Sah et al. [37] who showed that the concentration gradient does not influence encapsulated drug release until the onset of polymer degradation. In previous work by others using a double-emulsion method, a relatively high drug loading produced a particulate dispersion characterized by increased microparticle porosity rather than the desired molecular dispersion [36,38,39]. This underscores our present finding that higher ketamine loading led to a greater burst release. Another explanation is that some ketamine may have precipitated on the surface of the polymer rather than being incorporated into the polymer matrix [25], causing a relatively high apparent burst drug release. 

Our present data show that the use of this supercritical CO_2_ approach with the ester end-capped PLGA5050-1.5E resulted in microparticles with superior burst and sustained-release properties when compared with those produced using a range of PLGAs and PLAs either individually or in various blends. This could be due to the low glass transition temperature of PLGA5050-1.5E that enabled drug-loaded particles to be formed relatively quickly (*T*_g_ = 28.3 °C). Additionally, when the temperature of the scCO_2_ process is above that of the polymer *T*_g_, this allows higher CO_2_ sorption and swelling to enhance diffusion of the CO_2_/ketamine solution into the polymer matrix [40]. Microporosity of the PLGA5050-1.5E-based microparticles is likely to be the major factor aiding the diffusion of ketamine out of the CriticalMix microparticles, causing burst release as the larger area exposed for hydration facilitates rapid diffusion [41]. This is despite the fact that PLGA5050-1.5E has a relatively low inherent viscosity, a property conducive to the formation of particles with smaller pores or without pores. This topic has not been adequately addressed in published studies to date. 

Our present data also show that hydrophobic ester end-capped PLGAs are superior at controlling release compared with acid end-capped PLGAs, with the latter showing high burst ketamine release at ~55%, with 100% release within one week (Figure 3) but without adversely affecting the drug incorporation efficiency (Table 2). Our findings for ketamine are in contrast to work by others [36,42] using a classical emulsion method to microencapsulate large molecules (dextran and bovine serum albumin, respectively), whereby the hydrophilic acid end-capped PLGA RG503H (PLGA5050, MW 24–38 kDa, IV 0.32–0.44 dL/g, acid end-capped) led to a high drug loading and sustained release (~4 weeks). For the more hydrophobic RG502 (PLGA5050, MW 7–17 kDa, IV 0.16–0.24 dL/g, ester end-capped), these large molecules were trapped in the matrix and kept in the solid state because of the delayed hydration process [36]. In the current study, it is likely that the delayed hydration process of ester end-capped PLGAs [43] influenced the rate of polymer hydration in vitro and subsequently facilitated the release of ketamine out of the microparticles. This is in agreement with the findings by others that the encapsulation of weakly basic drugs into PLGA can accelerate the biodegradation of PLGA [44,45]. Additionally, the acids may also produce local lowering of the pH [35] to accelerate ketamine release. Therefore, to ensure that the release of the free base forms of small weakly basic molecules, such as ketamine, can be controlled for a longer period, acid end-capped polymers should be avoided. 

For ketamine-release kinetics, the high correlation coefficients for the Higuchi and Korsmeyer–Peppas models suggest a Fickian diffusion mechanism for most formulations [30,31,32]. The lowest burst release profile of F6 (10% DL, PLGA5050-1.5E based) fitted well to the zero-order model but not well to the Korsmeyer–Peppas model (*R*^2^ = 0.8330, *n* = 0.4824), suggesting that Fickian diffusion may play a critical role in the burst release. 

Previous work by others has demonstrated that the drug release profile could be tailored by altering the molar ratio of the different MW PLGA polymers used in the depot formulation [46]. It was found that PLGA blends generally had intermediate properties when compared with pure polymer formulations [46], suggesting that blends of low-, medium-, and high-MW polymers may form dense and less porous microparticles, enabling a better control of sustained-release properties. However, our data show that burst release was not improved as anticipated for these blended formulations. An alternative approach is to include excipients which swell on contact with water, such as poloxamer 407 (Koliphor F407) [47,48], with the aim of filling the exposed microparticle pores and slow the rate of encapsulated payload diffusion. Thus, polymers/excipients, which become molten at physiological temperature closing up the pores and thereby reducing the overall surface area exposed [49], should be beneficial for extended-release formulations [50]. Hence, future work aimed at the investigation of combinations of PLGA5050-1.5E with various swelling agents to reduce ketamine burst release may be beneficial in further increasing ketamine loading without compromising the desired zero-order sustained release profile. 

## 5. Conclusions

The preparation of ketamine-encapsulated microparticles was successfully achieved using supercritical fluid polymer encapsulation with EE in the range 60–100% in the absence of surfactants and potentially toxic organic solvents. A sustained-release profile extending for ≥21 days was achieved using ester end-capped PLGA5050-1.5E at drug loading of 10% in vitro. It is clear from our findings that payload solubility in scCO_2_ as well as CO_2_ sorption and swelling of the polymer vary with temperature, pressure, contact time, stirring rate of the scCO_2_/polymer/drug solution, as well as depressurization conditions including CO_2_ venting rate and venting temperature. By systematically varying these parameters and by selecting PLGAs with different physicochemical properties (end groups, viscosity, etc.), small molecule-loaded microparticles with the desired release rate can be produced using supercritical fluid polymer encapsulation instead of organic solvents and emulsifiers that are difficult and costly to remove.

## Figures and Tables

**Figure 1 pharmaceutics-10-00264-f001:**
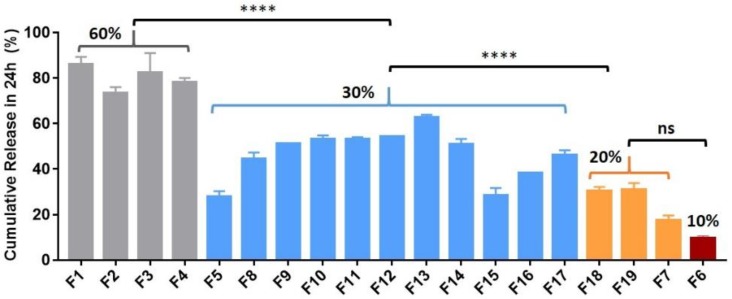
Mean (±SEM) in vitro ketamine percentage of cumulative release in 24 h for a range of formulations (*n* = 3). Reducing the ketamine payload from 60% to 30% and from 30% to 10–20% reduced the burst effect (****, *p* < 0.0001). One-way ANOVA followed by the Bonferroni test.

**Figure 2 pharmaceutics-10-00264-f002:**
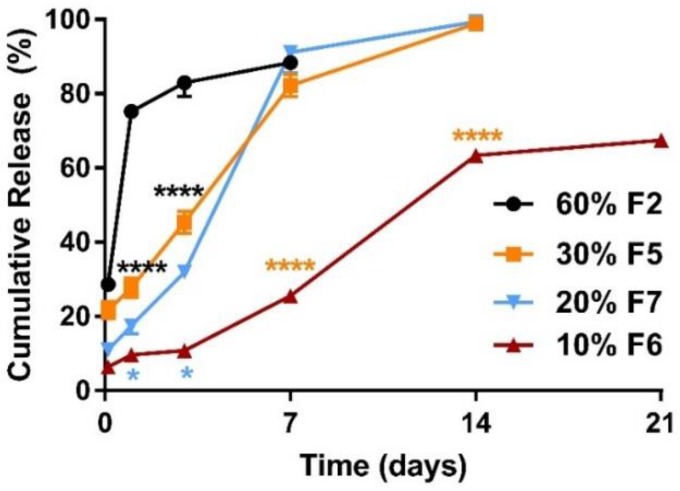
Comparison of mean (±SEM) in vitro ketamine release profiles between microparticle formulations (*n* = 3) with ketamine loading at 60% and 30%, 20% and 10%. The 24 h burst release decreased significantly (****, *p* < 0.0001) from 74.6% (● F2, 60% ketamine and 40% PLGA5050-1.5E) to 27.7% (■ F5, 30% ketamine and 70% PLGA5050-1.5E). Reducing the ketamine load from 30% to 20% and 10% using the same polymer, PLGA5050-1.5E, enabled the polymer matrix to have better control over ketamine release. Sustained release continued for 14 days when ketamine loading was 30% (■ F5), whereas it lasted more than 21 days at 10% ketamine loading (▼ F6). The 24 h burst release was reduced significantly from 27.7% to 9.0% (*, *p* < 0.05) for microparticles at 10% ketamine loading when compared with 30% ketamine loading. Ketamine loading at 20% (▲) and 30% (■) did not differ significantly (*p* > 0.05) in terms of release duration, but burst release was reduced from 27.7% to 17.5%. Two-way ANOVA followed by the Bonferroni test. F, Formulation.

**Figure 3 pharmaceutics-10-00264-f003:**
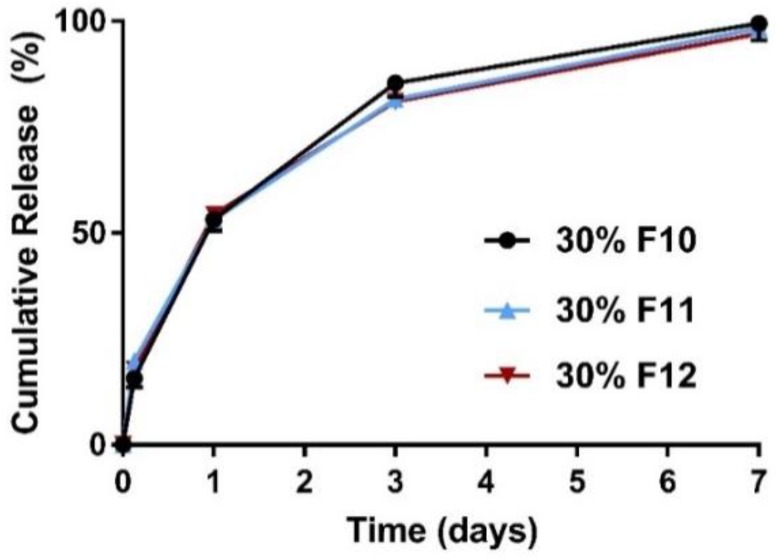
Mean (±SEM) in vitro ketamine release profiles did not differ significantly (*p* > 0.05) between the three acid end-capped polymers assessed (*n* = 3). A maximum of one-week sustained release was achieved in vitro. The cumulative release in 24 h was ~55%, >80% in three days, and 100% in one week for formulations 10–12, respectively. F, Formulation; F10, PLGA5050-1A (50%) and PLGA7525-4A (20%); F11, PLGA5050-2A (70%); F12, PLGA5050-1A (70%).

**Figure 4 pharmaceutics-10-00264-f004:**
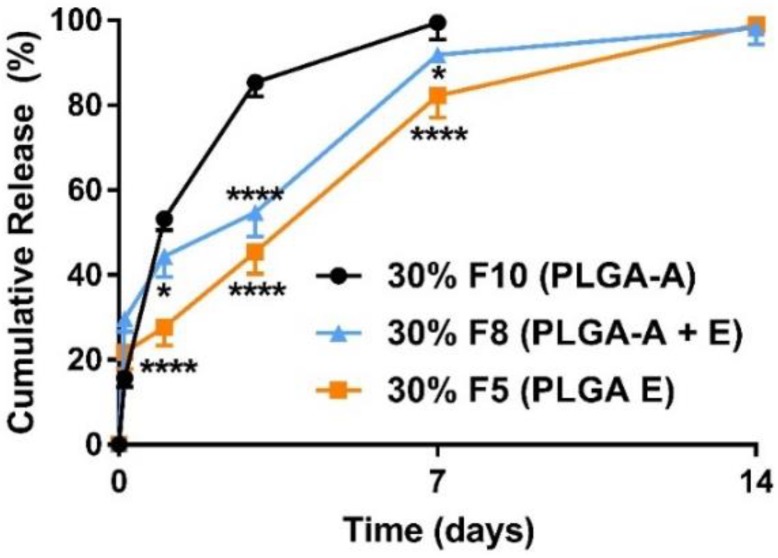
Mean (±SEM) in vitro ketamine release profiles varied for polymers (*n* = 3) with different end groups. Incorporation of a low-MW ester end-capped PLGA (▲ F8) with acid end-capped PLGA reduced the rate of release significantly (*, *p* < 0.05; ****, *p* < 0.0001) when compared with a blend of acid end-capped PLGAs (● F10, 50% of PLGA5050-1A and 20% of PLGA7525-4A). The 24 h burst release decreased from 53.2% for F10 to 44.4% for F8, and 100% release was achieved over seven days for both formulations. The low-MW ester end-capped PLGA itself improved the release profile (■ F5) compared with a blend of acid end-capped PLGAs (F10). The 24 h burst release decreased significantly (****, *p* < 0.0001) from 53.2% for F10 to 33.2% for F5, and 100% of ketamine was released in two weeks. Two-way ANOVA followed by the Bonferroni tests. F, Formulation.

**Figure 5 pharmaceutics-10-00264-f005:**
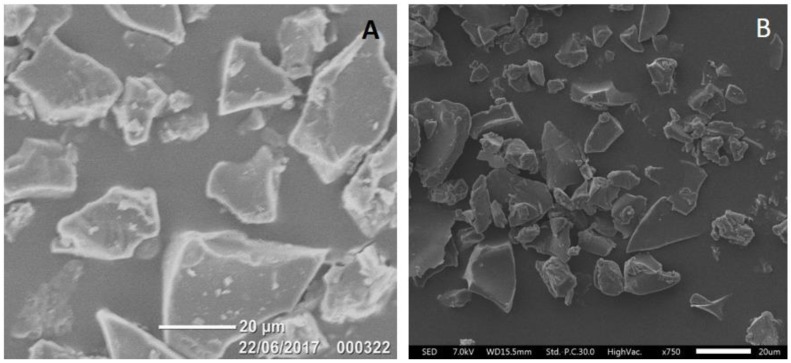
SEM micrographs illustrating the morphology and size of sustained-release ketamine-loaded microparticles (Formulation 6: 10% Ketamine + 90% PLGA5050-1.5E) stored at 2–8 °C for one month (**A**) and 6 months (**B**).

**Table 1 pharmaceutics-10-00264-t001:** Physicochemical properties of the PLGA and PLA polymers.

Polymer No.	Polymer Commercial Name	Composition	Inherent Viscosity (dL/g)	End Group
1	PLGA5050DLG-1.5E	PLGA 50:50	~0.2	Ester
2	RG504	PLGA 50:50	0.45–0.60	Ester
3	RG503	PLGA 50:50	0.32–0.44	Ester
4	RG752S	PLGA 75:25	0.8–1.2	Ester
5	PLGA9010DLG-4.5E	PLGA 90:10	0.4–0.5	Ester
6	R202S	PLA	0.16–0.24	Ester
7	PLA-2.5A	PLA	0.2–0.3	Acid
8	PLGA5050DLG-1A	PLGA 50:50	0.05–0.15	Acid
9	PLGA5050DLG-2A	PLGA 50:50	0.15–0.25	Acid
10	PLGA7525DLG-4A	PLGA 75:25	0.35–0.45	Acid

PLA, poly(lactic-acid); PLGA, poly(lactic-*co*-glycolic acid).

**Table 2 pharmaceutics-10-00264-t002:** Ketamine-loaded microparticles—formulations and incorporation efficiency.

Formulation (F)	Polymer Ratio (%)		Theoretical Loading (%)	Actual Loading (%)	Encapsulation Efficiency (EE %)
F1	PLGA-RG504 (20%)	PLA-2.5A (20%)	60	60 (±0.4)	100
F2	PLGA5050-1.5E (40%)		60	58 (±0.4)	97
F3	PLGA5050-1.5E (25%)	PLGA 7525-4A (15%)	60	54 (±2.2)	90
F4	PLA-2.5A (25%)	PLGA 7525-4A (15%)	60	58 (±9.7)	97
F5	PLGA5050-1.5E (70%)		30	26 (±1.6)	87
F6	PLGA5050-1.5E (90%)		10	9 (±0.8)	90
F7	PLGA5050-1.5E (80%)		20	16 (±0.6)	80
F8	PLGA5050-1.5E (50%)	PLGA7525-4A (20%)	30	30 (±2.6)	100
F9	PLGA5050-1.5E (23.3%)	RG503 (23.3%) RG752S (23.3%)	30	21 (±0.3)	70
F10	PLGA5050-1A (50%)	PLGA7525-4A (20%)	30	30 (±1.0)	100
F11	PLGA5050-2A (70%)		30	30 (±1.4)	100
F12	PLGA5050-1A (70%)		30	30 (±0.3)	100
F13	PLGA5050-1A (35%)	PLGA7525-4A (35%)	30	30 (±0.3)	100
F14	PLGA RG503 (35%)	PLGA RG752S (35%)	30	23 (±1.2)	77
F15	PLGA RG503 (20%)	PLGA RG752S (50%)	30	19 (±1.4)	63
F16	RG752S (40%)	PLGA9010-4.5E (30%)	30	19 (±1.6)	63
F17	PLGA RG752S (50%)	PLA R202S (20%)	30	30 (±0.4)	100
F18	RG503 (23%)	RG752S (57%)	20	18 (±0.3)	88
F19	PLGA RG752S (50%)	PLA R202S (20%)	20	20 (±1.4)	100

A, Acid end-capped; E, Ester end-capped.

**Table 3 pharmaceutics-10-00264-t003:** Fit parameters of the kinetic model for the release of ketamine-loaded PLGA5050-1.5E-based microparticles.

Kinetic Model	Zero Order		Higuchi		Korsmeyer–Peppas			
Equation	Mt = K_0_t Mt/M∞ = K_0_t		M_t_ = k_H_t^1/2^ Mt/M∞ = k_H_t^1/2^		M_t_/M∞ = k_KP_ t*^n^* log (M_t_/M∞) = log K_KP_ + *n* × log t			
Mechanism	Case II Transport		Fickian Diffusion		Depends on *n* Value, Shape and Material			
Formulation	Slope	*R* ^2^	Slope	*R* ^2^	Log K_KP_	K_KP_	*n*	*R* ^2^
Ketamine-loaded PLGA5050-1.5E-based microparticles with different theoretical DL (Figure 2)								
F6 (10%)	0.0023	0.9481	0.4070	0.9140	−0.4457	0.640	0.4824	0.8330
F7 (20%)	0.0051	0.8735	0.7608	0.9274	−0.1998	0.819	0.5037	0.9028
F5 (30%)	0.0045	0.8807	0.6891	0.9726	0.4900	1.633	0.3396	0.9041
F2 (60%)	0.0068	0.5405	0.8525	0.7918	0.8657	2.377	0.2861	0.8824
Ketamine-loaded microparticles with different polymers at 30% theoretical loading								
F10	0.0090	0.7758	1.0347	0.9521	0.1669	1.182	0.4737	0.9707
F11	0.0086	0.7848	0.9934	0.9591	0.3984	1.489	0.4085	0.9849
F12	0.0086	0.7728	0.9892	0.9526	0.3100	1.363	0.4322	0.9742
F14	0.0020	0.6643	0.4130	0.8559	0.8738	2.396	0.2355	0.9497
F15	0.0014	0.8026	0.3114	0.9331	0.7634	2.146	0.2271	0.9006
F17	0.0020	0.8292	0.4294	0.9251	0.8703	2.388	0.2344	0.9120
Reference	[30]				[31,32]			

DL, drug loading; K_0_ is the zero-order release constant; K_H_ is the Higuchi dissolution constant; K_kp_ is the Korsmeyer release rate constant; M_t_/M∞ is a fraction of drug released at time t; M_t_ is the amount of drug released in time t; M∞ is the amount of drug released after time ∞; *n* is the diffusional exponent or drug release exponent.

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
