# Peer review of "Formulation of Bioerodible Ketamine Microparticles as an Analgesic Adjuvant Treatment Produced by Supercritical Fluid Polymer Encapsulation"

_pharmaceutics, 2018, doi:10.3390/pharmaceutics10040264_

Round 1

Reviewer 1 Report

The manuscript by Han et al. describes formulation and in vitro evaluation of microparticles containing ketamine, with the long-term goal of providing sustained release for pain treatment. The use of supercritical CO2 as solvent is promising in terms of suitability for future pharmaceutical formulations, and the extensive release study is adequate to well characterize these microparticles. Overall, the manuscript is worth publication provided that these minor issues are addressed:

1) It is not clear the exact size and PDI of the microparticles. While I agree that conventional approaches (e.g. DLS) are not suitable for these large systems, it would be useful to provide at least a semiquantitative evaluation based on alternative techniques. This would better corroborate other found results and allow clear comparison with what already reported.

2) Dependence of release from the acid or ester terminal group is interesting. As the authors correctly pointed out, ketamine is a weakly basic compound, so it could be hypothesized an acid/salt equilibrium that could affect both loading efficiency and drug leakage (burst release). Ideally, the authors should comment on the possible effect of using weakly acidic drugs in terms of loading efficiency and release. Even better, they could provide an example of what happens in a specific case with weakly acidic drugs. This would surely provide useful guidelines for future design of drug-loaded microparticles for sustained release.

Author Response

Thank you.

1)    It is not clear the exact size and PDI of the microparticles. While I agree that conventional approaches (e.g. DLS) are not suitable for these large systems, it would be useful to provide at least a semiquantitative evaluation based on alternative techniques. This would better corroborate other found results and allow clear comparison with what already reported.

Thank you. We will implement this suggestion in work beyond the scope of this current manuscript.

2) Dependence of release from the acid or ester terminal group is interesting. As the authors correctly pointed out, ketamine is a weakly basic compound, so it could be hypothesized an acid/salt equilibrium that could affect both loading efficiency and drug leakage (burst release). Ideally, the authors should comment on the possible effect of using weakly acidic drugs in terms of loading efficiency and release. Even better, they could provide an example of what happens in a specific case with weakly acidic drugs. This would surely provide useful guidelines for future design of drug-loaded microparticles for sustained release.

On Page 11 line 338-339, this has been addressed. These two references has been added and the original references of 44-48 also updated to 46-50. All changes made to page 11 line 349, 351, 355, and 358; and line 490-503 on page 14 are shown in red font.

44.     Cha Y, Pitt CG 1989. The Acceleration of Degradation-Controlled Drug Delivery from Polyester Microspheres. Journal of Controlled Release  8(3):259-265.

45.           D'Souza S, Faraj JA, Dorati R, DeLuca PP 2015. Enhanced Degradation of Lactide-co-Glycolide Polymer with Basic Nucleophilic Drugs. Advances in Pharmaceutics  2015:10.

Reviewer 2 Report

The manuscript entitled “Formulation of Bioerodable Ketamine Microparticles as an Analgesic Adjuvant Treatment Produced by Supercritical Fluid Polymer Encapsulation” by Han et al describes the preparation and in vitro release profile of ketamine loaded PLGA microparticles. This is a well-designed study and interesting to the readers of Pharmaceutics. My major concerns are as follows:

1.       The formulations look like cake instead of microparticles (Fig 5). Is there any specific reason to call these as microparticles?

2.       To describe the burst release, author should perform release of selected formulations at early time points such as 0.5h and 1 h.

Author Response

Thank you.

1.        The formulations look like cake instead of microparticles (Fig 5). Is there any specific reason to call these as microparticles?

       The particles are random in shape with micro-size. For this reason, we call them microparticles.

2.        To describe the burst release, author should perform release of selected formulations at early time points such as 0.5h and 1 h.

In future work beyond the scope of this manuscript, we will check burst release at these early time points. The time points we used in our present are similar to those used in our previous study (Han et al J Pharm Sci 2015). We have checked formulations in our previous study at 1, 2, 3, 4, 5, and 6 hours.

Round 2

Reviewer 2 Report

The authors have adequately addressed my concerns and it could be accepted for publication in Pharmaceutics.